# Interpretable Knowledge Tracing with Multiscale State Representation

Submission Id: 355*

## ABSTRACT

Knowledge Tracing (KT) is vital for education, continuously monitoring students' knowledge states (mastery of knowledge) as they interact with online education materials. Despite significant advancements in deep learning-based KT models, existing approaches often struggle to strike the right balance in granularity, leading to either overly coarse or excessively fine tracing and representation of students' knowledge states, thereby limiting their performance. Additionally, achieving a high-performing model while ensuring interpretability presents a challenge. Therefore, in this paper, we propose a novel approach called Multiscale-state-based Interpretable Knowledge Tracing (MIKT). Specifically, MIKT traces students' knowledge states on two scales: a coarse-grained representation to trace students' domain knowledge state, and a fine-grained representation to monitor their conceptual knowledge state. Furthermore, the classical psychological measurement model, IRT (Item Response Theory), is introduced to explain the prediction process of MIKT, enhancing its interpretability without sacrificing performance. Additionally, we extended the Rasch representation method to effectively handle scenarios where questions are associated with multiple concepts, making it more applicable to real-world situations. We extensively compare MIKT with 20 state-of-the-art KT models on four widely-used public datasets. Experimental results demonstrate that MIKT outperforms other models while maintaining its interpretability. Moreover, experimental observations have revealed that our proposed extended Rasch representation method not only benefits MIKT but also significantly improves the performance of other KT baseline models. The code can be found on the anonymous website https://anonymous.4open.science/r/MIKT-BC12.

## CCS CONCEPTS

• **Applied computing** → **Education**; **Distance learning**.

## KEYWORDS

knowledge tracing, knowledge state representation, interpretable, educational data mining

## 1 INTRODUCTION

In the context of the web, where online education platforms like Coursera, MOOC, and others have become increasingly prevalent, Knowledge Tracing (KT) emerges as a critical topic. With the vast array of educational resources available online, it has become a crucial task for online learning platforms to model the learning process of their users, and further provide their users a personalized learning guidance.as they navigate through web-based learning environments[44]. KT plays a pivotal role in achieving this by continuously monitoring and assessing students' knowledge states based on their interactions with online educational materials[3].

In recent years, KT models based on deep learning have shown outstanding performance[10, 18, 25, 45]. Given that KT's central task is to trace students' knowledge states, designing an effective encoding for representing knowledge states becomes of paramount importance. Additionally, deep learning-based KT models often lack interpretability, which undoubtedly impedes the further application of KT.

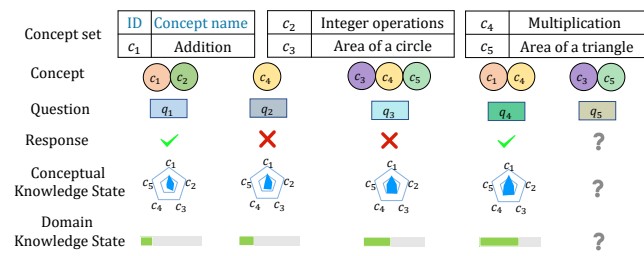

**Figure 1: A simple example about knowledge tracing**

We have presented a simple example of knowledge tracing in Figure 1, where each question is associated with one or more concepts. Students practice on different questions, gradually increasing their knowledge during the learning process. To represent the student's knowledge state, the graph-based KT model[21, 34] traces the student's knowledge states on all concepts (referred to as conceptual knowledge state in this paper) using hidden vectors and propagates these knowledge states through the specific graph structure. On the other hand, the sequential-based KT models[11, 27] do not trace the student's knowledge state on each concept. Instead, they directly model the student's current knowledge state (referred to as domain knowledge state in this paper) based on the student's historical interactions and represent this knowledge state using a hidden vector. These two modeling approaches are vastly different but have both achieved impressive success. It inevitably leads us to wonder: can we combine these two methods to better trace students' knowledge state? From a modeling perspective, the graph-based KT model can finely trace a student's knowledge state on the current question, while the sequential-based KT model provides a more coarse-grained representation of the student's overall knowledge state. We believe that both coarse-grained and fine-grained tracing of students' knowledge state is necessary. For example, suppose a student has recently answered multiple questions incorrectly. Now, when the student encounters a question that they are actually good at, the graph-based KT model might predict that the student will answer correctly because it assumes the student has a strong grasp of the underlying concept. On the other hand, the sequential-based KT model might predict that the student will answer incorrectly due to the student's recent history of consistently wrong responses. It might perceive that the student's learning state has been poor

lately, and therefore, even when the student encounters a question they should know, the model may still predict an incorrect response. Integrating both modeling approaches could potentially lead to a more comprehensive and accurate KT model. By leveraging the strengths of both fine-grained and coarse-grained tracing, we can better understand and predict students' learning progress.

Therefore, we believe that combining the two mentioned modeling approaches is a promising idea. However, several challenges need to be addressed. First, to the best of our knowledge, we are the first to propose the integration of graph-based KT models and sequential-based KT models. Effectively combining these two approaches to leverage their respective strengths remains an unexplored endeavor. How can we integrate these two modeling approaches to harness their potential? Second, sparse interactions between students and questions make it difficult for the model to accurately trace the students' knowledge states[31]. To address this issue, we need to consider how to represent the questions. This is a common problem faced by most KT models. Although AKT[10] introduced the Rasch[26] representation method, which significantly improved model performance and benefited other KT models, AKT assumes that a question is only related to a single concept, which does not align with real-world scenarios. Lastly, deep learning-based KT models often lack interpretability. The opaque decision-making process of these models limits their further applications. On the other hand, KT models that consider interpretability typically sacrifice performance, creating a trade-off between high performance and interpretability[5]. Therefore, designing a high-performance model with interpretability is a critical challenge.

In this paper, we propose a novel approach called Multiscale-state-based Interpretable Knowledge Tracing (MIKT) to address the challenges mentioned above. Specifically, MIKT trace students' knowledge states on two scales: a coarse-grained representation to monitor students' domain knowledge state and a fine-grained representation to monitor their conceptual knowledge state. The domain knowledge state is constructed based on the student's entire historical interactions, taking into account forgetting behavior at each time step. The conceptual knowledge state is built from the student's answers to questions related to specific concepts, considering the student's domain knowledge state and the time interval since their last responses for each concept to account for forgetting behavior. Since each question involves multiple concepts, an attention mechanism is used to aggregate conceptual knowledge states with different weights. When combining these two knowledge states, MIKT utilizes an attention mechanism to differentiate their respective roles. Additionally, MIKT extends the Rasch model to handle questions covering multiple concepts, aggregating them using attention while considering their difficulty levels. To ensure interpretability, MIKT not only relies on attention mechanisms but also introduces an IRT prediction module to explain model predictions. MIKT fits well with the IRT[38] prediction module since it can derive student abilities from multiple states and question difficulty using the expanded Rasch module, both of which are required by the IRT prediction module. This enhances the model's interpretability while maintaining performance. We extensively compare MIKT with 20 state-of-the-art KT models on four widely-used public datasets. Experimental results demonstrate that MIKT exhibits excellent performance and high interpretability.

This paper proposes a novel model called Multiscale-state-based Interpretable Knowledge Tracing (MIKT), and its contributions can be summarized as follows:

- To the best of our knowledge, we are the first to consider combining students' knowledge states from multiple scales, providing a new perspective for better tracing students' knowledge states.
- We fully utilize attention mechanisms to facilitate the transparent modeling process of MIKT. Additionally, an IRT prediction module is designed to interpret MIKT's output, seamlessly complementing MIKT and enhancing interpretability without compromising performance.
- We extend the Rasch model, which not only benefits MIKT but also significantly improves the performance of other KT baseline models.
- Through experiments on four widely-used public datasets and comparison with 20 state-of-the-art KT models, the results demonstrate that MIKT exhibits excellent performance in predicting student performance. Additionally, interpretability experiments confirm the high level of explainability achieved by MIKT.

## 2 RELATED WORK

### 2.1 Knowledge State Modeling for Deep Knowledge Tracing

Based on existing knowledge tracing models, we classify the mechanisms for representing student knowledge states into four distinct categories:

- **State-Sequential**: Models like DKT[25], ATKT[11], DIMKT[27], etc., are representative sequential-based KT models that trace students' knowledge states at discrete time steps.
- **State-Latent**: Represented by models such as DKVMN[45], SKVMN[1], DGMN[2], etc., these KT models assume that all questions are related to their predefined latent concepts and trace students' knowledge states on these latent concepts.
- **State-Graph**: Models like GKT[21], SKT[34], HGKT[33], etc., fall into this category. They are graph-based KT models that trace students' knowledge states on individual concepts and propagate them within the graph.
- **State-Free**: SAKT[23], AKT[10], DTransformer[44], sparseKT[12], etc., are leading attention-based KT models in this category. They do not trace students' knowledge states at each discrete time step; instead, they dynamically construct the students' current knowledge states based on their historical interactions.

### 2.2 Interpretable Deep Knowledge Tracing

Recently, an increasing number of interpretable methods have been adopted in KT models. These models can be broadly categorized into three types:

- **Post-hoc local explanation**: Aims to explain why the model makes certain predictions or decisions, such as [19] using LRP[4] techniques to propagate relevance scores from

the model's output layer to the input layer for interpreting KT models.

- **Globally interpretable with an interpretable structure**: Aims to design an interpretable model structure to understand the process of modeling knowledge states. For instance, [36] embeds an interpretable cognitive framework in the model to understand students' knowledge state modeling process.
- **Globally interpretable with interpretable parameters**: Aims to directly utilize interpretable parameters derived from the model to explain its predictions. For example, QIKT[5] exports three interpretable ability scores of students and uses them to make predictions in the output layer.

From the perspective of knowledge state modeling, our proposed MIKT does not fall into any specific category. Instead, MIKT combines aspects from both the first and third categories of modeling approaches, providing a novel perspective for tracing students' knowledge states. Experimental results have demonstrated the effectiveness of this modeling method. Regarding interpretability, MIKT belongs to the third category. However, unlike existing methods, MIKT is not solely based on attention mechanisms. It also relies on the advantages of tracing multiscale knowledge states in students and extending the Rasch representation method. This enables MIKT to derive students' ability values and question difficulty values in perfect alignment with the parameters needed for IRT predictions. Consequently, MIKT maintains a high level of interpretability while delivering exceptional performance.

## 3 MIKT FRAMEWORK

In this chapter, we will provide a detailed introduction to MIKT, and its overall architecture is illustrated in Figure 2. First, we present the problem formulation of KT. Subsequently, we provide a comprehensive overview of each module in MIKT: the Expanded Rasch Module, the Cognitive Thinking Module, the IRT Prediction Module, and the Cognitive Update Module. Finally, we outline the training process of the model.

### 3.1 Problem Formulation

The KT task can be formulated as follows: Given a student's historical interaction sequence represented as $X = (q_1, Neibor_{q_1}, a_1), (q_2, Neibor_{q_2}, a_2), ..., (q_t, Neibor_{q_t}, a_t)$, where $q_t$ denotes the question answered by the student at time $t$, $Neibor_{q_t}$ represents the set of related concepts to $q_t$ and $a_t \in \{0, 1\}$ indicates whether the student's response to the question is incorrect ($a_t = 0$) or correct ($a_t = 1$). The task of KT is to predict the probability $p(a_{t+1} = 1 | X, q_{t+1})$ of a student answering the next question $q_{t+1}$ correctly.

### 3.2 Expanded Rasch Module

Our proposed Expanded Rasch Module is capable of handling scenarios where a question is associated with multiple concepts. Specifically, given the original question embedding matrix $K \in R^{n \times d}$ and the concept embedding matrix $C \in R^{m \times d}$, where $n$ denotes the total number of questions, $d$ represents the embedding dimension, and $m$ indicates the total number of concepts, we use $K_i$ to represent

the $i$-th row of $K$ and $C_i$ to represent the $i$-th row of $C$. For the question $q_t$ at the current moment, in order to distinguish the effects of different concepts on the question, the concept representation $MC_{q_t}$ contained in the question $q_t$ is computed using the following equation:

$$MC_{q_t} = \sum_{j \in Neibor_{q_t}} qa_j * C_j$$

$$qa_j = \frac{exp(\frac{K_{q_t}^T C_j}{sqrt(d)})}{\sum_{i \in Neibor_{q_t}} exp(\frac{K_{q_t}^T C_i}{sqrt(d)})} \qquad (1)$$

Among these, $Neibor_{q_t}$ represents the set of concepts related to question $q_t$. Furthermore, inspired by Rasch[26] theory, we should pay attention to the characteristics inherent in the question itself, such as its difficulty level and how it differs from the associated concepts. Specifically, the features $OF_{q_t}$ encompassed in the question $q_t$ are obtained through the following equation:

$$OF_{q_t} = diff_{q_t} * (W_1 \sum_{j \in Neibor_{q_t}} \frac{1}{|Neibor_{q_t}|} C_j + b_1) \qquad (2)$$

Among these, $diff_{q_t}$ represents the difficulty level of question $q_t$, which is a scalar. $W_1$ and $b_1$ are learnable parameters, and $|Neibor_{q_t}|$ denotes the total number of concepts associated with question $q_t$. Note that the calculation here takes the average of related concepts, mainly because we thought the degree of change of the question is compared to the concepts it is associated with.

Finally, assuming the final question embedding matrix is represented as $Q \in R^{n \times d}$, where $Q_i$ denotes the $i$-th row of $Q$, then:

$$Q_{q_t} = MC_{q_t} + OF_{q_t} \qquad (3)$$

### 3.3 Cognitive Thinking Module

MIKT traces students' domain knowledge state $H \in R^{T \times d}$ and conceptual knowledge state $HS \in R^{m \times d}$, where $T$ represents the total number of time steps. $H_i$ and $HS_i$ represent the $i$-th row of $H$ and $HS$, respectively. At the time step $t$:

We denote the student's current domain knowledge state as $\widetilde{H_t}$. It is essential to note that the student's knowledge state is not constant but gradually fades over time, considering the forgetting behavior. For the student's domain knowledge state at the previous time step $H_{t-1}$, if the time interval since the last interaction is represented as $It_t^H$ after embedding, the domain knowledge state of the student after forgetting can be calculated as follows:

$$\widetilde{H_t} = H_{t-1} * \sigma(\beta(H_{t-1} \oplus It_t^H)) \qquad (4)$$

Among them, $\sigma$ represents the Sigmoid function, $\beta$ represents non-linear transformation, and $\oplus$ represents concatenation operation.

For the concept set $Neibor_{q_t}$ related to the current question $q_t$, MIKT trace students' knowledge states on these concepts. Specifically, for any concept $j \in Neibor_{q_t}$, MIKT obtain the student's knowledge state on concept $j$ denoted as $HS_j$ based on its index. MIKT also consider the forgetting behavior for the conceptual knowledge state, and denote the time interval since the student's last response to concept $j$ as $It_j^{HS}$, which is represented through embedding. Thus, the student's conceptual knowledge state is obtained

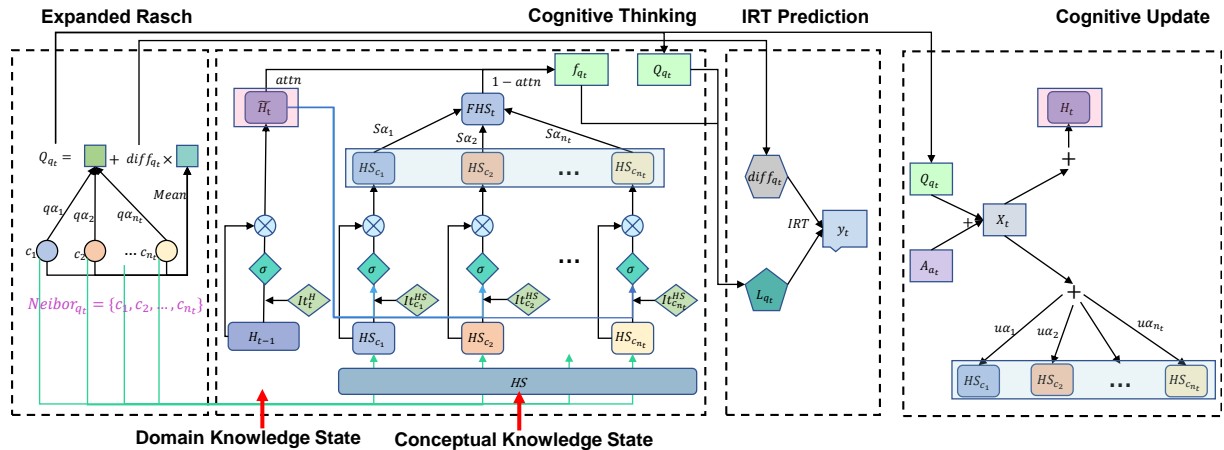

**Figure 2: The overall framework of MIKT. Firstly, the final question embeddings are obtained by utilizing the Expanded Rasch module. In the Cognitive Thinking module, the forgotten representations of Domain Knowledge State and Conceptual Knowledge State are obtained, respectively, and then aggregated. Predictions are made using the IRT Prediction module. In the Cognitive Update module, the Domain Knowledge State and Conceptual Knowledge State used in the Cognitive Thinking module are updated based on the student's responses, respectively.**

after considering forgetting behavior:

$$HS_j = HS_j * \sigma(\xi(HS_j \oplus It_j^{HS} \oplus \widetilde{H_t})) \tag{5}$$

Please note that the clarification here is that the time interval in this context is the interval in terms of time steps. For instance, if a student's last response for concept $j$ was at time step $\alpha$, then the time interval from the current moment is $t - \alpha$. This approach offers two advantages: firstly, it simplifies calculations, and secondly, it eliminates the need to introduce additional time-related features, which may not be present in all datasets. Furthermore, the non-linear transformation is represented by $\xi$. Different notations for non-linear transformations represent different learnable parameters. It is worth noting that during the computation of $HS_j$, we incorporate $\widetilde{H_t}$. This consideration arises from acknowledging that a student's domain knowledge state may impact their conceptual knowledge state. Consequently, MIKT aggregate the student's knowledge states for all relevant concepts to derive the student's current conceptual knowledge state at time $t$, denoted as $FHS_t$:

$$FHS_t = \sum_{j \in Neibor_{q_t}} Sa_j * HS_j$$

$$Sa_j = \frac{exp(\frac{Q_{q_t}^T HS_j}{sqrt(d)})}{\sum_{i \in Neibor_{q_t}} exp(\frac{Q_{q_t}^T HS_i}{sqrt(d)})} \tag{6}$$

In order to derive the student's ultimate knowledge state $f_{q_t}$, we employ varying levels of attention to $\widetilde{H_t}$ and $FHS_t$:

$$f_{q_t} = attn * \widetilde{H_t} \oplus (1 - attn) * FHS_t$$

$$attn = \sigma(\pi(Q_{q_t} \oplus \widetilde{H_t} \oplus FHS_t)) \tag{7}$$

Where $\pi$ represents non-linear transformation, $attn$ is an intermediate variable that represents the student's attention to the domain

knowledge state $\widetilde{H_t}$. Clearly, $1 - attn$ indicates the student's attention to the conceptual knowledge state $FHS_t$. It's worth noting that concatenation is used here instead of addition, considering the different roles played by various levels of knowledge states.

### 3.4 IRT Prediction Module

In order to increase the transparency of MIKT's final decision-making process, an IRT prediction module has been implemented to provide explanations for MIKT's predictions. This module utilizes the student's current knowledge state $f_{q_t}$ and the question $Q_{q_t}$ to calculate the student's ability value $L_{q_t}$ in relation to the specific question:

$$L_{q_t} = \sigma(\varphi(f_{q_t} \oplus Q_{q_t})) \tag{8}$$

$\varphi$ represents non-linear transformation. Afterwards, we extract the current question's difficulty value $diff_{q_t}$ from the expanded Rasch module. Following the principles of the IRT[38] method, a student's prediction is influenced by their ability value and the question's difficulty value. MIKT's prediction $y_t$ is determined through the following equation:

$$y_t = \sigma(5 * (L_{q_t} - diff_{q_t})) \tag{9}$$

Please be aware that the multiplication by 5 serves the main purpose of achieving smoother output values for the Sigmoid function and does not hold any other particular significance.

### 3.5 Cognitive Update Module

After students complete their answers, the knowledge will be updated according to their performance, and the acquired knowledge will be represented as $X_t$:

$$X_t = Q_{q_t} + A_{a_t} \tag{10}$$

Here, $A \in R^{2 \times d}$ denotes the answer embedding, and $A_i$ represent the $i$-th row of $A$. We use the following equation to obtain the

student's final domain knowledge state $H_t$ at the moment $t$ after the process of learning and acquisition:

$$H_t = \widetilde{H_t} + Tanh(\psi(X_t)) \tag{11}$$

The Tanh function is denoted as $Tanh$, and $\psi$ represents non-linear transformation. The gain in conceptual knowledge state be represented as:

$$G_t = Tanh(\omega(X_t)) \tag{12}$$

Here, $\omega$ denotes non-linear transformation. For the knowledge state of any relevant concept $j \in Neibor_{q_t}$ at the current time step, it is updated using the following equation:

$$HS_j = HS_j + ua_j * G_t$$

$$ua_j = \frac{exp(\frac{G_t^T HS_j}{sqrt(d)})}{\sum_{i \in Neibor_{q_t}} exp(\frac{G_t^T HS_i}{sqrt(d)})} \tag{13}$$

## 3.6 Model Training

The KT loss $Loss_{KT}$ is defined as the binary cross-entropy loss between the prediction $y_t$ and the true answer $a_t$, and it is computed as follows:

$$Loss_{KT} = -\sum_{i=1}^{T}(a_t log y_t + (1-a_t)log(1-y_t)) \tag{14}$$

We utilize the Adam[13] algorithm to optimize the model parameters.

## 4 EXPERIMENT

In this section, we conducted extensive experiments to answer the following questions:

- **RQ1**: How does MIKT perform?
- **RQ2**: How do different components of MIKT impact its performance? Is the proposed multiscale-state tracing beneficial?
- **RQ3**: How does MIKT provide explanations for its predictions?
- **RQ4**: Can the expanded Rasch representation method proposed by MIKT improve the performance of other knowledge tracing baseline models?

## 4.1 Experimental Setting

*4.1.1 Datasets.* We evaluated the performance of MIKT on four commonly used public datasets: ASSIST09, ASSIST12, EdNet and Eedi. The specific introduction and processing methods of the dataset can be found in Appendix A, and the statistical information of the dataset is presented in Table 1.

**Table 1: Summary statistics of processed datasets.**

|  | ASSIST09 | ASSIST12 | EdNet | Eedi |
|---|---|---|---|---|
| # Student | 4,160 | 5,000 | 5,000 | 5,000 |
| # Question | 15,680 | 36,056 | 11,775 | 26,706 |
| # Concept | 167 | 242 | 1,837 | 1,050 |
| # Interaction | 207,659 | 717,188 | 1,156,254 | 597,124 |

*4.1.2 Baseline Model.* To assess the performance of MIKT, we compared it with 20 state-of-the-art KT models as follows:

- **DKT**[25]: Traces students' knowledge states using LSTM.
- **DKT+**[43]: Enhances DKT by addressing inconsistent knowledge states and irrecoverable inputs.
- **KQN**[14]: Predicts students' performance using knowledge state encoder and concept encoder.
- **DKT+forgetting**[20]: Augments DKT by incorporating various behavioral features to consider forgetting in student knowledge states.
- **PEBG+DKT**[16]: Enhances DKT by deeply exploring the relationship between questions and concepts to obtain pretrained question representations.
- **GIKT**[41]: Use GCN to aggregate the relationship between questions and concepts to enhance question representation.
- **ATKT**[11]: Use adversarial training to improve the generalization ability of the model.
- **CDKT**[8]: On the basis of DKT, use contrastive learning between questions to represent questions.
- **DIMKT**[27]: Traces the impact of question difficulty on students' knowledge states.
- **QIKT**[5]: Models centered around questions and enhances model interpretability with interpretable parameters.
- **AT-DKT**[17]: Enhances DKT with two additional tasks to improve performance.
- **DKVMN**[45]: Traces students' knowledge states using a dynamic key-value memory network.
- **Deep-IRT**[42]: Integrates IRT with DKVMN to improve interpretability.
- **GKT**[21]: Propagates students' conceptual knowledge states using a graph structure.
- **SAKT**[23]: Captures student-concept relationships using self-attention mechanism.
- **SAINT**[6]: Fully employs a Transformer[35] architecture to model students' knowledge states.
- **AKT**[10]: Simulates students' forgetting behavior using context-based attention mechanism.
- **CL4KT**[15]: Mitigates sparsity in student-concept interactions using contrastive learning.
- **simpleKT**[18]: Simplifies the model structure based on AKT, achieving simplicity without sacrificing performance.
- **DTransformer**[44]: Uses contrastive learning to trace students' stable knowledge states.

It is worth noting that all the KT models compared to MIKT share the same input configuration, which includes only the question, concept, and response as input features (this is also the most common input setup in KT research). KT models, such as DGMN[2], SKT[34], LPKT[28], and LBKT[40], that require additional input features were not included in the comparison because it could lead to an unfair performance comparison.

*4.1.3 Implementation Details.* We implemented MIKT using PyTorch[24] with the following settings: the learning rate was set to 0.002, the batch size was 80, and the embedding dimension was 64. Additionally, L2 weight regularization with a weight decay of 1e-5 was applied to the model's weights. To avoid the issue of gradient explosion, we consistently set the gradient clipping threshold to

**Table 2: Comparison of MIKT and 20 KT models on four datasets. Best results in bold, next best underlined. * indicates t-test p-value < 0.05 compared to the second best result. Model grouping details can be found in Section 2 (RELATED WORK).**

| Model | Model Group | Interpretable | ASSIST09 AUC | ASSIST09 ACC | ASSIST12 AUC | ASSIST12 ACC | EdNet AUC | EdNet ACC | Eedi AUC | Eedi ACC |
|---|---|---|---|---|---|---|---|---|---|---|
| DKT[25] | State-Sequential | × | 0.7684 | 0.7297 | 0.7328 | 0.7367 | 0.7006 | 0.7129 | 0.7629 | 0.7182 |
| DKT+[43] | State-Sequential | × | 0.7783 | 0.7337 | 0.7373 | 0.7350 | 0.7028 | 0.6698 | 0.7484 | 0.7079 |
| KQN[14] | State-Sequential | √ | 0.7546 | 0.7249 | 0.7230 | 0.7330 | 0.6909 | 0.7117 | 0.7583 | 0.7143 |
| DKT+forgetting[20] | State-Sequential | × | 0.7717 | 0.7295 | 0.7362 | 0.7359 | 0.7018 | 0.7159 | 0.7642 | 0.7186 |
| PEBG+DKT[16] | State-Sequential | × | 0.7738 | 0.7329 | 0.7518 | 0.7495 | 0.7571 | 0.7366 | 0.7853 | 0.7310 |
| GIKT[41] | State-Sequential | × | 0.7726 | 0.7301 | 0.7672 | 0.7506 | 0.7640 | 0.7366 | 0.7924 | 0.7362 |
| ATKT[11] | State-Sequential | × | 0.7735 | 0.7332 | 0.7347 | 0.7363 | 0.7027 | 0.7109 | 0.7663 | 0.7195 |
| CDKT[8] | State-Sequential | × | 0.7733 | 0.7297 | 0.7720 | 0.7547 | 0.7645 | 0.7386 | 0.7920 | 0.7360 |
| DIMKT[27] | State-Sequential | × | 0.7704 | 0.7310 | 0.7621 | 0.7484 | 0.7623 | 0.7368 | 0.7908 | 0.7338 |
| QIKT[5] | State-Sequential | √ | 0.7801 | 0.7377 | 0.7707 | 0.7529 | 0.7579 | 0.7327 | 0.7932 | 0.7363 |
| AT-DKT[17] | State-Sequential | × | 0.7671 | 0.7293 | 0.7425 | 0.7405 | 0.7039 | 0.7136 | 0.7649 | 0.7180 |
| DKVMN[45] | State-Latent | × | 0.7629 | 0.7266 | 0.7228 | 0.7329 | 0.6975 | 0.7120 | 0.7590 | 0.7162 |
| Deep-IRT[42] | State-Latent | √ | 0.7657 | 0.7279 | 0.7253 | 0.7345 | 0.6997 | 0.7124 | 0.7609 | 0.7173 |
| GKT[21] | State-Graph | × | 0.7666 | 0.7290 | 0.7261 | 0.7333 | 0.6943 | 0.7104 | 0.7618 | 0.7170 |
| SAKT[23] | State-Free | × | 0.7564 | 0.7192 | 0.7296 | 0.7348 | 0.6956 | 0.7115 | 0.7556 | 0.7123 |
| SAINT[6] | State-Free | × | 0.7515 | 0.7134 | 0.7643 | 0.7477 | 0.7621 | 0.7370 | 0.7866 | 0.7293 |
| AKT[10] | State-Free | × | 0.7850 | 0.7429 | 0.7830 | 0.7599 | 0.7647 | 0.7385 | 0.7882 | 0.7340 |
| CL4KT[15] | State-Free | × | 0.7626 | 0.7275 | 0.7236 | 0.7331 | 0.6965 | 0.7118 | 0.7583 | 0.7147 |
| simpleKT[18] | State-Free | × | 0.7772 | 0.7315 | 0.7786 | 0.7571 | 0.7627 | 0.7373 | 0.7885 | 0.7307 |
| DTransformer[44] | State-Free | × | 0.7646 | 0.7223 | 0.7672 | 0.7515 | 0.7501 | 0.6954 | 0.7531 | 0.7315 |
| MIKT | State-Sequential+State-Graph | √ | **0.7938***| **0.7454***| **0.7834** | **0.7608** | **0.7703***| **0.7430***| **0.7954***| **0.7392***|

15.0 during training. For data preprocessing, we removed sequences with a length less than 3 from the dataset. Since the input sequence lengths varied, we uniformly set all sequences to a fixed length of 200. For each dataset, we used 80% of all sequences as the training set and 20% as the test set[41, 44]. We conducted the experiment five times and reported the average results[2, 32]. All models are trained on a Linux server with two 2.00GHz Intel(R) Xeon(R) CPUs and a Nvidia Tesla P100-PCIE-16GB GPU. Consistent with prior work[22, 29, 30, 39, 46], we use AUC (Area Under the Curve) as the first evaluation metric and ACC (Accuracy) as the second metric. The greater their values, the better the model's performance.

## 4.2 Performance (RQ1)

*4.2.1 Overall Performance.* Table 2 presents the performance comparison between MIKT and other KT models, with the best results highlighted in bold and the second-best results underlined. According to Table 2, we can observe the following: (1) MIKT demonstrates superior performance across all datasets, underscoring the effectiveness of the proposed method in this paper. (2) When comparing MIKT to State-Sequential and State-Graph KT models, MIKT exhibits a noticeable performance improvement, indicating the effectiveness of constructing students' knowledge states from multiple perspectives. (3) Comparing Deep-IRT to DKVMN in State-Latent, Deep-IRT enhances model interpretability through the introduction of IRT methods while slightly improving performance, suggesting that IRT methods do not significantly sacrifice KT model performance. (4) Among the State-Free KT models, AKT stands out as a robust baseline, owing to its two key components: representing questions using the Rasch method and employing a context-aware attention structure to mimic student forgetting behavior. This underscores the importance of question representation and forgetting in KT. (5) Among all compared baseline models, none achieve both

high performance and interpretability. In contrast, MIKT maintains high performance while also offering interpretability.

*4.2.2 T + N Prediction.* To better simulate real student question-answering scenarios and evaluate the stability of MIKT performance, we conducted a $T + N$ prediction experiment. Specifically, in this experiment, we not only predicted the performance of students in answering questions at the next moment $T + 1$ (assuming the current moment is $T$) but also predicted their performance at moments $T + 2, T + 3, ..., T + N$. As shown in Figures 3 and 4, we compared MIKT with some high-performing KT models (see Table 2) on all datasets and made the following observations: (1) As the value of $N$ in $T + N$ increased, the performance of all models generally exhibited a decreasing trend, which is expected because KT models primarily predict performance at $T + 1$. However, in all cases, MIKT showed the slowest decrease in performance. This demonstrates the advantage of MIKT, namely, its ability to trace students' knowledge states across multiple scales, which proves to be more stable compared to tracing knowledge states at a single scale. (2) We found that AKT exhibited unstable predictive performance in this experiment. This may be because State-Free KT methods construct knowledge states based on specific questions as queries, and when there is a need to predict multiple different questions, these knowledge states may not be highly relevant to the questions that need to be answered. (3) MIKT consistently maintained high performance in all situations and consistently outperformed other KT models. This indicates that MIKT's ability to trace knowledge states is both stable and effective.

## 4.3 Ablation Studies (RQ2)

Four variants of MIKT were constructed to explore the impact of different components on MIKT, as shown in Table 3. Specifically, "w/o ERM" removes the Expanded Rasch Module, "w/o Forget"

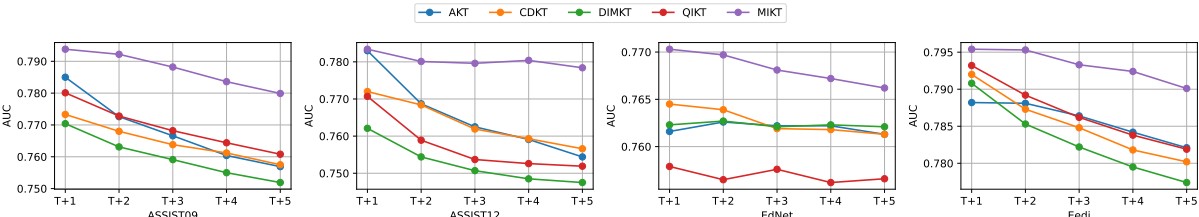

**Figure 3: Comparing T+N prediction performance (AUC) of MIKT and well-performing KT models across four datasets.**

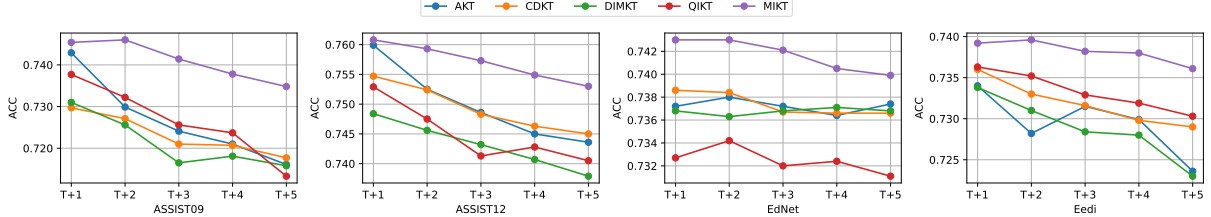

**Figure 4: Comparing T+N prediction performance (ACC) of MIKT and well-performing KT models across four datasets.**

**Table 3: Performance comparison of ablation study.**

| Model | ASSIST09 | | ASSIST12 | | EdNet | | Eedi | |
|---|---|---|---|---|---|---|---|---|
| | AUC | ACC | AUC | ACC | AUC | ACC | AUC | ACC |
| MIKT w/o ERM | 0.7822 | 0.7371 | 0.7787 | 0.7577 | 0.7669 | 0.7415 | 0.7925 | 0.7378 |
| MIKT w/o Forget | 0.7780 | 0.7314 | 0.7657 | 0.7504 | 0.7649 | 0.7392 | 0.7829 | 0.7296 |
| MIKT w/o DKS | 0.7723 | 0.7280 | 0.7658 | 0.7498 | 0.7535 | 0.7322 | 0.7488 | 0.7076 |
| MIKT w/o CKS | 0.7868 | 0.7412 | 0.7756 | 0.7571 | 0.7658 | 0.7395 | 0.7924 | 0.7368 |
| MIKT | **0.7938** | **0.7454** | **0.7834** | **0.7608** | **0.7703** | **0.7430** | **0.7954** | **0.7392** |

disregards the Forgetting of knowledge states, "w/o DKS" eliminates Domain Knowledge States, and "w/o CKS" omits Conceptual Knowledge States. The following observations were made: (1) "w/o ERM" exhibited a similar degree of performance decline across all datasets, with the most noticeable decrease observed in ASSIST09. This underscores the importance of question representation. It's worth noting that, in comparison to other modules, the reduction in performance with "w/o ERM" is not particularly pronounced. This is not because question representation is less important but rather because it is not as crucial for MIKT due to its excellent knowledge state tracing capabilities. MIKT does not heavily rely on question representation. In subsequent experiments, we will demonstrate how ERM-based question representation significantly enhances the performance of some simpler KT models. (2) "w/o Forget" experienced an average performance decrease of around 1%, indicating the significant role of forgetting in the student's knowledge state. This aligns with intuition, as knowledge forgetting is an essential aspect of the learning process. (3) "w/o DKS" and "w/o CKS" both exhibited varying degrees of performance decrease, with "w/o DKS" showing the most significant decline. This is in line with expectations since building "DKS" involves the entire interaction history of the student, whereas constructing "CKS" only relies on relevant interaction history. Furthermore, the combination of both

"DKS" and "CKS" significantly improved their respective performance, underscoring the effectiveness of the multi-scale approach proposed in this paper.

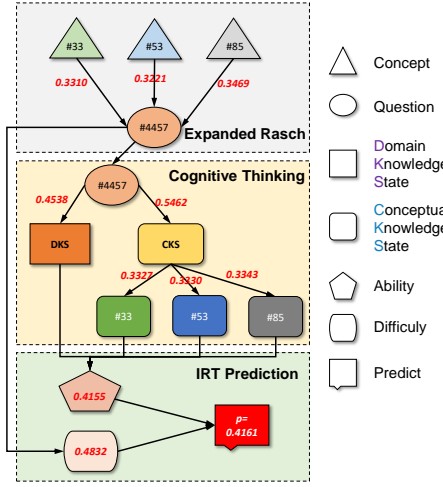

**Figure 5: The interpretable prediction process of MIKT.**

## 4.4 Explainable thinking process (RQ3)

To demonstrate the interpretability of MIKT, we randomly selected a student from the ASSIST09 dataset and observed MIKT's prediction process for the student's response to question 4457 and its related concepts 33, 53, and 85, as shown in Figure 5. First, in the Expanded Rasch module, MIKT estimated the relevance of question 4457 to these concepts as 0.3310, 0.3221, and 0.3469, respectively, and calculated the difficulty of the question as 0.4832. Next, in the Cognitive Thinking module, MIKT predicted the student's attention

**Table 4: Enhance other KT baseline models with MIKT's ERM and compare with Rasch method using AUC.**

|  | DKT | DKT+Rasch | DKT+ERM | DKVMN | DKVMN+Rasch | DKVMN+ERM | SAKT | SAKT+Rasch | SAKT+ERM |
|---|---|---|---|---|---|---|---|---|---|
| ASSIST09 | 0.7684 | 0.7827 | **0.7865** | 0.7629 | 0.7751 | **0.7812** | 0.7564 | 0.7715 | **0.7742** |
| ASSIST12 | 0.7328 | 0.7700 | **0.7719** | 0.7228 | **0.7698** | 0.7690 | 0.7296 | **0.7723** | 0.7665 |
| EdNet | 0.7006 | 0.7626 | **0.7650** | 0.6975 | 0.7608 | **0.7644** | 0.6956 | 0.7626 | **0.7639** |
| Eedi | 0.7629 | 0.7942 | **0.7968** | 0.7590 | 0.7900 | **0.7941** | 0.7556 | 0.7862 | **0.7889** |

to domain knowledge state as 0.4538 and the overall attention to conceptual knowledge state as 0.5462. Within conceptual knowledge states, MIKT predicted the student's attention to each relevant conceptual knowledge state as 0.3327, 0.3330, and 0.3343, respectively. Finally, in the IRT prediction module, MIKT combined the student's current knowledge state with the question to calculate the student's ability value for that question as 0.4155. Considering the question's difficulty of 0.4832, MIKT predicted the probability of the student answering the question correctly as 0.4161. Clearly, MIKT's decision process is fully transparent and interpretable. In the context of this example, if the student were to answer the question incorrectly, MIKT could provide the following explanations: (1) The student's ability value is relatively low, which may be due to the student's overall lower proficiency or insufficient mastery of the specific concept/question. (2) The question is relatively challenging for this student. Intuitively, the student's ability value is lower than the question's difficulty, making it difficult for the student to answer correctly. In a real-world scenario, if a teacher had access to information about a student's attention to these knowledge states, they could provide targeted assistance. For instance, if it became evident that the key to solving question 4457 is a deep understanding of concept 85, and the teacher noticed that the student's attention to concept 85's knowledge state was insufficient (e.g., in this case, only 0.5462 * 0.3343 = 0.1826), the teacher could intervene and advise the student to pay more attention to the knowledge state related to concept 85. As a result, the student's ability value for the question would significantly improve (due to having a clearer direction for solving it), increasing the probability of answering it correctly.

### 4.5 Multi-concept Rasch extension (RQ4)

We use the proposed ERM module to enhance some simple and commonly used KT models, including DKT, DKVMN, and SAKT, and compare their performance with the Rasch[26] representation method proposed by AKT[10], as shown in Table 4. The approach taken here is the same as AKT's, which means modifying their question representations to ERM's question representation. It can be observed that: (1) Whether using the Rasch method or the ERM method, both significantly improve the performance of these simple KT models. This undoubtedly underscores the importance of question representation. (2) On the ASSIST12 dataset, the ERM method does not consistently outperform the Rasch method. This may be due to the fact that each question in the ASSIST12 dataset is associated with only one concept, and Equation 2 applies a linear transformation to the average related concepts, potentially limiting the expressive power of question variations. (3) In most cases, using the ERM method performs better than using the Rasch method to enhance other models. This indicates the importance of considering

questions in relation to multiple related concepts, as opposed to Rasch, which only considers a single concept. The ERM method not only enhances model performance but also aligns better with real-world scenarios.

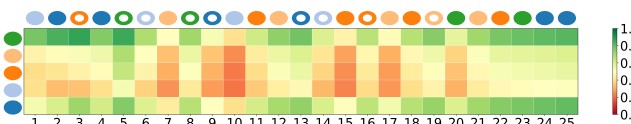

**Figure 6: Knowledge tracing along a learning sequence.**

## 5 APPLICATION

The most interesting application of KT may be to trace the knowledge state of students. It allows us to gain a better understanding of students' learning progress. In Figure 6, we showed the evolution process of tracing a student's knowledge state using MIKT. Specifically, we randomly selected a student from the ASSIST09 dataset who solved five questions at 25 time steps, we distinguish each question with a different colored circle. Show the corresponding circle for correct answers, and add a white circle on the respective circle for wrong answers. As shown, when the student answers a question correctly/incorrectly during practice, their mastery of the question clearly strengthens/weakens. Moreover, as time goes by, we find that the student gradually forget questions he/she have previously mastered. In addition, as the number of practice questions increases, the traced knowledge state becomes more stable. Overall, the student's final knowledge state significantly improves compared to the beginning, even though he/she answered some questions incorrectly. This is consistent with our intuition that practicing questions benefits students' knowledge mastery, and even answering questions incorrectly can enhance knowledge.

## 6 CONCLUSION

In this paper, we propose a novel model called Multiscale-state-based Interpretable Knowledge Tracing (MIKT). It aims to trace students' domain knowledge state and conceptual knowledge state, and the results demonstrate its remarkable effectiveness. Additionally, we extend the Rasch representation method, benefiting not only MIKT but also significantly enhancing the performance of other knowledge tracing baseline models. Furthermore, MIKT introduces IRT approach to improve model interpretability. Experimental results on four commonly used public datasets show that MIKT outperforms current state-of-the-art knowledge tracing models while maintaining high interpretability.

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

## A  DATASETS

We evaluated the performance of MIKT on four commonly used public datasets:

- **ASSIST09**[9][1]: Collected from the ASSISTments online educational platform during 2009-2010.
- **ASSIST12**[9][2]: Collected from the ASSISTments online educational platform during 2012-2013.
- **EdNet**[7][3]: A dataset collected by Santa[7], an online tutoring platform, from 2017 to 2019.
- **Eedi**[37][4]: Used for the NeurIPS 2020 Education Data Mining Challenge, collected by the online education platform Eedi from 2018 to 2020.

Based on previous research, for the ASSIST series datasets, we removed scaffold questions and records without concepts[10]. Moreover, due to the large scale of ASSIST12, EdNet, and Eedi datasets, and limitations in computational resources, we randomly sampled records from 5000 students[41]. The statistical information for these datasets is provided in Table 1.

---

[1]https://sites.google.com/site/assistmentsdata/home/2009-2010-assistment-data
[2]https://sites.google.com/site/assistmentsdata/home/2012-13-school-data-with-affect
[3]https://github.com/riiid/ednet
[4]https://eedi.com/projects/neurips-education-challenge

