# OpenReview forum: "Interpretable Knowledge Tracing with Multiscale State Representation"
_ACM.org/TheWebConf/2024/Conference — TheWebConf24_

### Official Review · Reviewer_kgFt · 2023-11-18

**Novelty:** 5
**Technical Quality:** 5

**Review:**

Quality：
This paper introduces a Multiscale-state-based Interpretable Knowledge Tracing(MIKT) model, designed to enhance model interpretability while representing students’ knowledge states in both fine-grained and coarse-grained ways. The model comprises four components:
(1)	Expanded Rasch: Addresses the challenge of representing situations where a single question corresponds to multiple concepts.
(2)	Cognitive Thinking: Combines ideas from both graph-based and sequential-based knowledge tracing(KT) models, leveraging the strengths of both approaches.
(3)	IRT Prediction: Enhances the transparency of the decision-making process by introducing Item Response Theory (IRT) to improve model interpretability.
(4)	Cognitive Update: Updates students’ knowledge states by considering student abilities and question difficulty values, thereby enhancing the model's effectiveness.
Furthermore, this paper conducts experiments in four main aspects: a comparison of the overall model performance, ablation experiments, verification of model interpretability, and assessing the impact of the extended Rasch module on other Knowledge Tracing (KT) models. This paper provides detailed descriptions of all the experimental results. For example, it explains the reasons behind the performance fluctuations of AKT in Section 4.2.2 and delves into the reasons why the extended Rasch module performs less effectively than the regular module in the RQ4 experiment. However, it's worth noting that This paper does not consider replacing the extended Rasch module with the Rasch module in ablation experiments to confirm the effectiveness of the extended Rasch module. Additionally, in the RQ4 experiment, it is evident that the Rasch module significantly enhances other models, while the extended Rasch module's improvement over models with the Rasch module is not as pronounced. Moreover, the RQ4 experiment solely employs AUC as an evaluation metric and does not present results in terms of accuracy (ACC), which might raise concerns about the overall persuasiveness of the findings.

Clarity：
There are some minor clarity issues in the content, as follows:
(1)	Line 54: In "learning guidance.as they ..." there is an extra period before "as". It should be removed.
(2)	Lines 197-198: Between "2 RELATED WORK" and "2.1 Knowledge State Modeling for Deep Knowledge Tracing," it is suggested to insert a paragraph summarizing related work. This would help readers understand the content of related work right from the beginning.
(3)	It is recommended that the authors briefly introduce the shortcomings of the prior methods mentioned in the related work section and explain how the MIKT model improves upon those deficiencies.
(4)	Lines 202-203: "representing student knowledge states" should be changed to "representing students’ knowledge states" to maintain consistency with other parts of the text.
(5)	In equations 1, 6, and 13, explanations for qa_{j}, Sa_{j}, and ua_{j} are recommended.
(6)	For equation 2, it is not clear how the diff_{qt} values for each question are obtained. This should be detailed in This paper.
(7)	For equation 9, it is unclear why a subtraction symbol "－" is used in the calculation. It should be briefly explained whether this is related to the principles of IRT.
(8)	Equation 11 is not well represented in Figure 2. It is recommended that the authors update the figure to better reflect the equation.
(9)	In Figure 2, it is unclear how Q_{qt} in the Expanded Rasch Module differs from Q_{qt} in the Cognitive Update Module. If they are the same, there should be consistency in how they are depicted. Also, the symbol diff_{qt} shows inconsistent formatting in the figure.
(10)	In Figure 2, it is unclear if qα_{1} in the Expanded Rasch Module, Sα_{1} in the Cognitive Thinking Module, and uα_{1} in the Cognitive Update Module represent the same concept as qa_{j}, Sa_{j}, and ua_{j} in the respective formulas. The difference in symbols (α vs. a) should be addressed and modified for consistency.
(11)	In Figure 2, the right boundary dashed line of the Cognitive Update Module has inconsistent line thickness compared to other dashed lines.
(12)	In Table 4, the use of only AUC as an evaluation metric is questioned. It is suggested to include ACC results to make the results more convincing.
(13)	It is unclear how the "knowledge state of students" in Section 5 and the student knowledge state shown in Figure 2 differ from the previously described "domain knowledge state" and "conceptual knowledge state." This paper should clarify this distinction.
(14)	In Figure 6, This paper does not explain why some students answered questions incorrectly but their knowledge state for that question was higher than in the previous time step. For instance, at time step 5, a student answered a green question incorrectly, but their knowledge state for that question is higher than in the previous step.
(15)	The conclusion lacks information about the limitations of the MIKT model and the author's future work. This should be included.
(16)	In lines 1222-1223, it is suggested that the authors provide a detailed explanation of the random sampling method used for selecting 5,000 student records in the appendix.
These suggestions aim to enhance the clarity, completeness, and overall quality of This paper.


Originality：
(1)	This paper introduces a method for representing students' knowledge states at multiple scales, combining graph-based and sequential-based KT models to fuse fine-grained and coarse-grained representations of knowledge states. This paper is the first to propose the integration of graph-based and sequential-based KT models.
(2)	This paper introduces an IRT prediction module that attempts to explain the prediction process of MIKT by introducing interpretable parameters, enhancing the model's interpretability while maintaining prediction performance.
(3)	This paper extends the Rasch module, using an attention mechanism to aggregate the impact of multiple concepts on the same question, addressing the issue of representing questions corresponding to multiple concepts. This module is not only applicable to the MIKT method proposed in This paper but can also be applied to other commonly used KT models, improving model performance.
(4)	This paper uses a relatively simple method for combining attention weights of different components when using the attention mechanism. There is not a significant innovation in the calculation method. Additionally, the IRT method used has been applied in previous KT models without highlighting the differences from previous methods.


Significance：
This paper points out the challenge of balancing granularity in existing KT models. Therefore, it combines graph-based and sequential-based KT models for the first time and introduces an extended Rasch module to represent student knowledge states at both fine-grained and coarse-grained levels. Additionally, MIKT incorporates an IRT module in the prediction part to enhance the overall interpretability of the model. However, it is worth noting that the introduction of the IRT method in KT models is not entirely novel, as it has been applied in previous KT models. This paper does not emphasize how the use of IRT in MIKT differs from previous methods, and when explaining the results with the IRT method, it does not provide a detailed background on IRT. This, to some extent, may diminish This paper's significance.

Pros:
(1)	The writing of this paper is standardized.
(2)	This paper introduces the MIKT model, which is a novel combination of graph-based and sequential-based KT models. It enhances the relationship and representation of concepts and questions through an improved Rasch module and introduces IRT for explaining the model's prediction process.
(3)	This paper has a clear research motivation, providing a comprehensive description of the model's construction process. It conducts extensive comparative experiments with 20 state-of-the-art KT models, offering substantial experimental evidence. Additionally, it reasonably explains observed phenomena in the experiments.
(4)	This paper provides reproducible information, such as the use of public datasets and open-sourcing the source code.

Cons:
(1)	Some symbols in This paper's formulas are not clearly defined.
(2)	There are inconsistencies in symbol usage and presentation between Figure 2 and the corresponding formulas in the text.
(3)	This paper's approach to attention is relatively simplistic and lacks substantial innovation.

**Questions:**

(1)	How does MIKT adapt to individual differences in learning styles and learning abilities among students?
(2)	How does MIKT handle noisy or incomplete data, such as when students skip questions or deliberately provide incorrect answers?
(3)	Could the author discuss potential applications of MIKT outside the field of educational data mining?
(4)	What are the limitations of MIKT, and how does the author plan to address them in future work?

**Reviewer Confidence:**

3: The reviewer is confident but not certain that the evaluation is correct

**Scope:**

4: The work is relevant to the Web and to the track, and is of broad interest to the community

---

### Official Review · Reviewer_QW4u · 2023-11-22

**Novelty:** 4
**Technical Quality:** 4

**Review:**

Strengths
1. Explores knowledge tracking across various granularities of learning.
2. Offers empirical evidence supporting its efficacy.
3. Provides a comparative analysis between graph-based and sequential-based Knowledge Tracing (KT) models.

**Questions:**

1. The methodology proposed in the paper is incremental and challenging to follow. Understanding the purpose and problem-solving aspects of each module constructed by the authors is unclear. Where does "Cognitive Thinking" specifically manifest, and how is continuous learning visualized? How does the introduction of new knowledge pose challenges, and how does the computation resolve them? What constitutes interpretable knowledge? Can the authors offer clearer definitions of domain knowledge and conceptual knowledge? A more detailed explanation is needed.
2. The title mentions "Interpretable Knowledge Tracing," yet the content lacks demonstrable interpretability. If the authors refer to the proposed "classical psychological measurement model, IRT (Item Response Theory)," it falls short. Moreover, the IRT Prediction Module is hard to follow.
3. While the paper mentions the challenge of achieving high-performing models with interpretability, it fails to address it. The code does not utilize innovative gradient backpropagation or parallel approaches to achieve the high performance outlined in the paper.
4. Writing conventions lack consistency, with inconsistencies in tense usage and unclear symbols. For instance, "extended" in L27 is in the past tense, while "compare" in L29 is in the present tense. The meaning of "w" in equations remains ambiguous.
5. The experiments are insufficiently detailed and lack clear setups, leading to conclusions without robust support. For instance, how are specific knowledge states eliminated in the ablation experiments? Can a decrease in accuracy (ACC) solely signify interpretability in representations? Statements like "due to its excellent knowledge state tracing capabilities. MIKT does not heavily rely on question representation" lack substantiation. Additionally, there's confusion arising from the mention of knowledge forgetting.

**Reviewer Confidence:**

3: The reviewer is confident but not certain that the evaluation is correct

**Scope:**

3: The work is somewhat relevant to the Web and to the track, and is of narrow interest to a sub-community

---

### Official Review · Reviewer_NGx4 · 2023-11-24

**Novelty:** 5
**Technical Quality:** 5

**Review:**

This paper introduces the MIKT model which represents the students’ knowledge state on two scales: a coarse-grained and a fine-grained representation to monitor students’ domain knowledge and conceptual knowledge respectively. The MIKT model also uses the Rasch representation method to enrich the questions representation and applies the IRT model to make the prediction process interpretable. As a result, the proposed model could guarantee the performance as well as the interpretability at the same time. The experimental results on several public datasets demonstrates the effectiveness of the MIKT model.
This paper has following strengths:
1. Combining the strengths of graph-based tracing and sequential-based tracing by representing the knowledge states on two scales.
2. The extended Rasch module and IRT prediction module improve MIKT’s interpretability.
3. The paper conducted a large amount of experiments and analyses across 20 baselines and 4 datasets, enhancing the proposed model’s credibility.

And the weaknesses are listed below:
1. The clarification in Explainable thinking process section is not so convincing which owes the student’s failure to his insufficient attention to the corresponding important concept.
2. Additional experiments are needed to show the interpretability of the trainable difficulty level parameter.

**Questions:**

I have the following concerns and I hope the authors could make some further illustrations:
Q1: In section 4.4, the authors owe the student’s failure to his insufficient attention to concept 85. However, I think the student’s performance on a specific question depends on his proficiency on the corresponding concepts. I hope the authors could clarify whether the attention score is correlated with the proficiency to some extent.
Q2: The trainable parameter $diff_{q_t}$ in equation(2) is quite important in both the Rasch module and IRT module, which is the key to MIKT ’s interpretability. I hope the authors can provide additional experiments to show whether the parameter could really express the relative difficulty level of different questions.

**Reviewer Confidence:**

4: The reviewer is certain that the evaluation is correct and very familiar with the relevant literature

**Scope:**

3: The work is somewhat relevant to the Web and to the track, and is of narrow interest to a sub-community

---

### Official Review · Reviewer_Rpte · 2023-11-24

**Novelty:** 4
**Technical Quality:** 3

**Review:**

The paper proposes a knowledge tracing algorithm which uses an expanded Rasch model and multiscale state representation (a domain knowledge state + a concept knowledge state). Experiment results show that the proposed algorithm performs better than 20 existing knowledge tracing models on four datasets.

One issue with the experiments is that there is no hyper-parameter tuning for baseline models. The same hyper-parameters recommended/used in the original paper/codes are used on different datasets in the experiments. Without proper hyper-parameter tuning, the reported small performance gain may not be real .

The datasets used in the experiments are very small. Some datasets have more interactions, but they are reduced to smaller datasets by randomly sampling only 5000 users. The proposed model seems hard for parallel computing, which raises concern on the efficiency of the proposed model.

On main claim of the paper is interpretability, but there is no comparison between MIKT and other interpretable KT models like QIKT and Deep-IRT in terms of interpretability. The interpretability of the model is demonstrated using two Figures:

 - In Figure 5, the predicted probability is decomposed into two knowledge states and some attention weights. I find that is not convincing as from the two knowledge states to the final prediction, there are still other model parameters, like non-linear transformation $\pi$ in equation 7 and $\varphi$ in equation 8. Selectively showing some intermediate model outputs are not sufficient to show the interpretability of the model. There is no evidence to support that the numbers shown in Figure 5 are better in any way.

- Figure 6 is confusing too. The student answered the first question (green) correctly twice out of 4 answers, and the third question (red) correctly 4 times out of 6 answers, but the model predicts that the student masters the first question most of the time (scores are high) and does not master the 3rd question most of the time (scores are low). Students' knowledge states or ability on a same question are not expected to change rapidly back and forth in real world. However, the predicted scores on a same problem (e.g., 3rd question) in Figure 6 change dramatically within just 25 interactions, which means the scores predicted by the proposed model is very unstable. This instability may pose challenges to interpreting the predicted scores meaningfully.


I acknowledge that I have read the rebuttals. In the rebuttals, the authors confirm that the proposed model is not easy for parallel computing because it is a State-Sequential+State-Graph method. After reading the rebuttals, I still think that the significance and contribution of the paper is quite limited for the following reasons:

- the improvement on accuracy is small: mean AUC on four datasets is 0.55% higher than AKT and no proper hyper-parameter tuning is done for AKT. With proper hyper-parameter tuning, the performance gain can be even smaller.

- claim on interpretability is not convincing and not supported by any evidence: both Figure 5 and Figure 6 used for demonstrating the interpretability of the model have some issues as mentioned above.

- efficiency is a concern: the proposed model is around 10 times slower than AKT per training epoch. The largest dataset used in the experiments has only around 1.2 million interactions, but it takes 3 hours to train the proposed model.

**Questions:**

How the hyper-parameters of baseline models are being tuned?

The proposed model seems hard for parallel computing. What is the running time of the model? How does its running time compare with other models like AKT?

Figure 6 is confusing. The student answered the first question (green) correctly twice out of 4 answers, and the third question (red) correctly 4 times out of 6 answers, but the model predicts that the student masters the first question most of the time (scores are high) and does not master the 3rd question most of the time (scores are low). Is this expected?

**Reviewer Confidence:**

4: The reviewer is certain that the evaluation is correct and very familiar with the relevant literature

**Scope:**

4: The work is relevant to the Web and to the track, and is of broad interest to the community

---

### Official Review · Reviewer_c3dj · 2023-11-25

**Novelty:** 5
**Technical Quality:** 5

**Review:**

The paper presents MIKT, a novel approach in the domain of Knowledge Tracing. The existing deep learning-based KT models struggle with granularity balance and interpretability. MIKT addresses these challenges by tracing knowledge states at two scales: a coarse-grained domain knowledge state and a fine-grained conceptual knowledge state by combining graph-based tracing and sequence-based tracing. It also incorporates Item Response Theory (IRT) for interpretability, extends the Rasch representation method for handling multi-concept questions, and demonstrates improved performance over 20 state-of-the-art KT models.

Strength:

1.	The paper clearly outlines the motivation to improve knowledge tracing at various levels of detail and the approach of integrating a graph-based model with a sequence-based model is a logical and well-founded method for this purpose.

2.	The added IRT and Rasche module improves the model explanablity

3.	The experimental results are extensive and demonstrate that the model outperforms over 20 leading knowledge tracing (KT) models, showcasing significant improvement in performance.

Weakness:

1.	The approach used in the paper is incremental. Combining attention weights from different module, while effective, is not a novel approach. This raises concerns about the paper's originality and contribution to the field beyond incremental improvements.

2.	The notation used throughout the paper is not uniform. Specifically, there is ambiguity in Figure 2 regarding the symbols qα_{1} in the Expanded Rasch Module, Sα_{1} in the Cognitive Thinking Module, and uα_{1}. This inconsistency makes it difficult to understand the relationship and differences between these terms.

**Questions:**

How does MIKT address varying learning patterns and potential irregularities in student interaction data? Additionally, has the author accounted for instances where the data may contain noise?

**Reviewer Confidence:**

3: The reviewer is confident but not certain that the evaluation is correct

**Scope:**

3: The work is somewhat relevant to the Web and to the track, and is of narrow interest to a sub-community

---

### Decision · Program_Chairs · 2024-01-22

**Decision:**

Accept

**Comment:**

This paper presents a solution named "Multiscale-state-based Interpretable Knowledge Tracing" (MIKT) to trace students' knowledge states on two scales: a coarse-grained representation to trace students' domain knowledge state, and a fine-grained representation to monitor their conceptual knowledge state. And the classical Item Response Theory is introduced to enhance MIKT's interpretability. Empirical evaluations demonstrate the effectiveness of the proposed solution.

 The reviewers appreciated the proposed multi-scale representation for knowledge tracking and acknowledged the advantageous performance of the proposed solution against a rich set of baselines. However, they also questioned the novelty of the proposed solution, which is considered as incremental. Another major concern fell onto the evaluation of the claimed explanability: only qualitative results were demonstrated, which cannot convincingly prove the solution's explanablity.

 Overall, the reviewers recognized the contribution of the proposed solution and thus recommended this paper for acceptance. But we do urge the authors to further polish the paper's content according to the reviewers' comments and suggestions.